# Akt1-Inhibitor of DNA binding2 is essential for growth cone formation and axon growth and promotes central nervous system axon regeneration

Hyo Rim Ko[1,2], Il-Sun Kwon[1,2], Inwoo Hwang[1,2], Eun-Ju Jin[1,2], Joo-Ho Shin[1,2], Angela M Brennan-Minnella[3], Raymond Swanson[3], Sung-Woo Cho[4], Kyung-Hoon Lee[2,5], Jee-Yin Ahn[1,2]*

[1]Department of Molecular Cell Biology, Sungkyunkwan University School of Medicine, Suwon, Republic of Korea; [2]Center for Molecular Medicine, Samsung Biomedical Research Institute, Sungkyunkwan University School of Medicine, Suwon, Republic of Korea; [3]The Department of Neurology, University of California, San Francisco Medical Center, San Francisco, United States; [4]Department of Biochemistry and Molecular Biology, University of Ulsan, College of Medicine, Seoul, Republic of Korea; [5]Department of Anatomy, Sungkyunkwan University School of Medicine, Suwon, Republic of Korea

*For correspondence: jeeahn@skku.edu

**Abstract** Mechanistic studies of axon growth during development are beneficial to the search for neuron-intrinsic regulators of axon regeneration. Here, we discovered that, in the developing neuron from rat, Akt signaling regulates axon growth and growth cone formation through phosphorylation of serine 14 (S14) on Inhibitor of DNA binding 2 (Id2). This enhances Id2 protein stability by means of escape from proteasomal degradation, and steers its localization to the growth cone, where Id2 interacts with radixin that is critical for growth cone formation. Knockdown of *Id2,* or abrogation of Id2 phosphorylation at S14, greatly impairs axon growth and the architecture of growth cone. Intriguingly, reinstatement of Akt/Id2 signaling after injury in mouse hippocampal slices redeemed growth promoting ability, leading to obvious axon regeneration. Our results suggest that Akt/Id2 signaling is a key module for growth cone formation and axon growth, and its augmentation plays a potential role in CNS axonal regeneration.

## Introduction

Developmental axon growth or axon regeneration requires active molecular machinery that regulates specific transcription factors, growth cone components, and mediators of signal transduction (*Fawcett, 2001*; *Tanabe et al., 2003*; *Raivich et al., 2004*; *Lasorella et al., 2006*). Injured axons of the adult central nervous system (CNS) do not regenerate, because the ability to activate growth genes and growth cone substantially declines as neurons mature (*Fawcett, 2001*; *Fernandes and Tetzlaff, 2001*), and the CNS environment is hostile to those processes (*Filbin, 2006*; *Goldberg et al., 2002*; *Silver and Miller, 2004*). Neutralization of environmental inhibition is not sufficient for axon regeneration; therefore, elucidating intrinsic growth capacity and regulation of the neuron after injury is of critical importance (*Yiu and He, 2006*; *Lee et al., 2009*; *Fawcett et al., 1992*). Indeed, recent studies have proposed that reactivation of the intrinsic growth ability promotes CNS axon regeneration (*Rossi et al., 2001*; *Teng and Tang, 2006*; *Bouquet and Nothias, 2007*; *Smith et al., 2009*; *Leibinger et al., 2013*; *Watkins et al., 2013*). It has been proposed that

the mechanisms involved in axonal regeneration of the mature CNS have many features in common with those important in CNS development (*Cui, 2006*; *Harel and Strittmatter, 2006*).

In addition to its role in neuronal survival (*Ahn et al., 2004b*; *Ahn and Ye, 2005*), Akt/PKB (protein kinase B) signaling controls a variety of neuronal responses. It regulates both axon establishment and elongation both during development and in the regeneration of mature neurons through glycogen synthase kinase 3 (GSK3). However, the mechanism of GSK3 control of peripheral axon regeneration is controversial and its function in CNS axon regeneration remains unknown (*Jiang et al., 2005*; *Yoshimura et al., 2006*; *Kim et al., 2011*; *Saijilafu et al., 2013*; *Zhang et al., 2014*; *Gobrecht et al., 2014*). Moreover, Akt links a host of signaling molecules through activation of mTORC1, which regulates cap-dependent protein translation by inhibiting TSC1/2 to allow axon development, growth, and regeneration in CNS (*Ma et al., 2008*; *Li et al., 2008*; *Morita and Sobue, 2009*; *Park et al., 2008*). However, some evidence suggested an mTORC1 independent pathway that regulates axon regrowth in phosphatase and tensin homolog (*Pten*) deficient neurons (*Park et al., 2008*; *Yang et al., 2014*), which causes aberrant activation of Akt signaling. Thus, although Akt signaling encompasses developmental regulation of the intrinsic neuronal growth and axon regeneration after injury, the roles and molecular mechanism of Akt signaling in the growth of CNS axons remain to be determined.

Inhibitor of DNA binding 2 (Id2) is a negative regulator of basic helix-loop-helix (bHLH) transcription factors. During development, Id2 binds to bHLH transcription factors and hampers their ability to activate transcription of several growth inhibitory molecules and receptors, thus promoting axon growth (*Jackson, 2006*). Id2 degradation by a complex of the anaphase-promoting complex/cyclosome and its activator Cdh1 (APC/C$^{Cdh1}$) reduces axonal growth in the adult (*Stegmüller and Bonni, 2005*; *Lasorella et al., 2006*). Conversely, protection from Id2 degradation results in erratic growth and an abnormal distribution of parallel fibers in the cerebral cortex (*Konishi et al., 2004*), while enhanced Id2 expression in the dorsal root ganglion (DRG) promotes axonal growth after spinal cord injury (*Yu et al., 2011*). Thus, Id2 contributes to axonal growth during development and may also be involved in the intrinsic inability of the injured axons to regenerate in the adult (*Lasorella et al., 2006*). However, to our knowledge the specific temporal and spatial signals that may regulate the molecular changes induced by Id2 are not yet understood.

In this study, we defined the role of Akt in regulating Id2 functions in axon growth during development and attempted to enhance Akt/Id2 signaling after injury to promote axon regeneration. We identified Id2, as a new binding partner and novel kinase substrate of Akt. Akt-mediated phosphorylation of serine 14 (S14) on Id2 augmented its protein stability through disruption of the association of Id2 and E3 ligase Cdh1. During neuronal differentiation, S14- phosphorylated Id2 is predominantly enriched in the growth cones at the axonal tips where it facilitates axonal growth. This contributed to the maintenance of the growth cone via interaction with radixin, one of the ezrin, radixin, and moesin (ERM) family of proteins, which links F-actin to the plasma membrane. Moreover, in organotypic hippocampal slice culture, reactivation of Akt/Id2 signaling by adeno-associated virus (AAV) two after injury, prominently increased regrowth of axons, whereas ablation of Akt-dependent phosphorylation of Id2 caused failure in axonal regeneration. Our study suggests the molecular basis of intrinsic growth regulation of Akt/Id2 signaling and delineates the potential role of Akt/Id2 in CNS axonal regeneration.

## Results

### Akt binds to Id2 and phosphorylates serine 14

Because of the multiple downstream effectors of Akt in both the neuronal soma and axon terminal, this pathway might coordinate different steps of axon growth during development. In an effort to identify downstream targets of the Akt signal that might be involved in the regulation of axon growth, we examined protein interaction profiles using proteomic analysis in PC12 cells stably transfected with a constitutively active (CA) form or a kinase-dead (KD) form of Akt. Interestingly, our proteomic analysis showed that Id2 is a potent binding partner of active Akt (*Figure 1—figure supplement 1*). Indeed, we found endogenous interaction between Akt and Id2 in mouse brain lysates (*Figure 1A*); the specific interaction was confirmed in mouse brain extract using purified glutathione S-transferase (GST)-Id2 protein (*Figure 1B*). Employing Flag-tagged Akt isoforms (Akt1-3), we

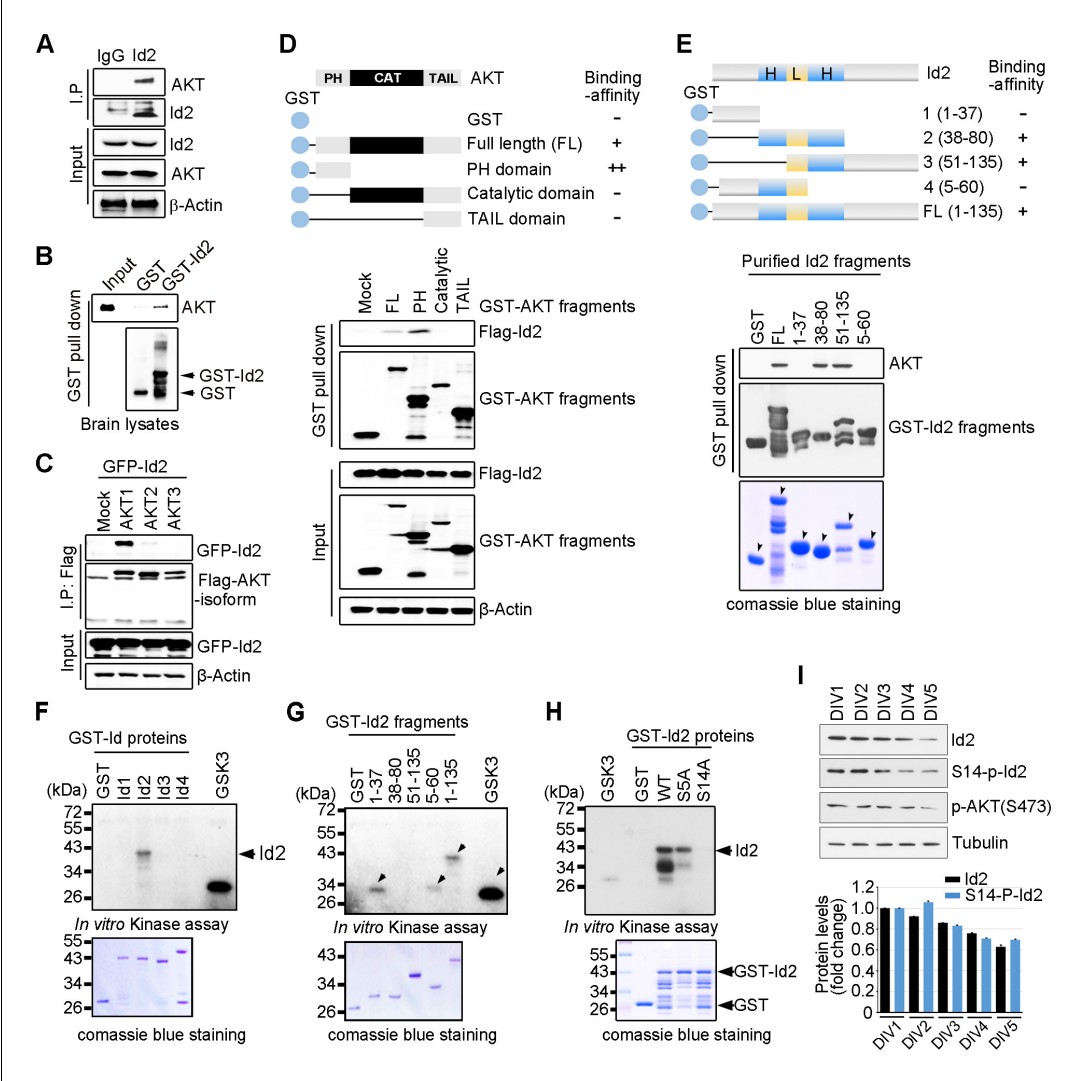

**Figure 1.** Akt binds to Id2 and phosphorylates Serine 14. (**A**) Mouse brain lysates were subjected to immunoprecipitation (IP)/immunoblotting (IB) with the indicated antibodies. (**B**) GST pull-down assays with purified GST-Id2 protein and P1 mouse brain lysates. (**C**) Flag-Akt1, 2, or three wee transfected into HEK293T cells together with GFP-Id2, and lysates were subjected to anti-Flag IP followed by IB as indicated. (**D**) Schematic diagram of the Akt fragments (upper). Flag-Id2 was co-transfected with mammalian GST-Akt fragments into 293T cells and lysates were subjected to GST pull-down assay and IB as indicated (bottom). (**E**) Schematic diagram of the Id2 fragments (upper). Purified GST-Id2 fragment proteins were pre-bound to GST-resin and reacted with lysate from PC12 cells followed by IB (bottom). Arrows indicate purified Id2 fragments protein. (**F–H**) In vitro Akt kinase assay was performed with purified GST-proteins and purified active Akt. GSK3β fusion and GST proteins were used as positive and negative controls, respectively. Arrows indicate purified Id2 fragments protein or phosphorylated GST-Id2 fragments proteins (**E–G**). (**I**) IB of DIV1-5 cortical neuron lysates probed on the indicated antibodies. Densitometry analysis of IB is shown in the bottom. Data are representative of at least three independent experiments. See also *Figure 1—figure supplements 1* and *2*.

The following figure supplements are available for figure 1:

**Figure supplement 1.** AKT interacts with Id2.

**Figure supplement 2.** Akt binds to Id2 and phosphorylates Serine 14 on Id2.

verified that, among the three isoforms evaluated, Id2 interacts with Akt1 (*Figure 1C*). Our mapping analysis showed that the PH domain of Akt interacts with Id2 (*Figure 1D*). Reciprocal experiments with a series of purified GST-tagged Id2 fragments demonstrated that the helix domain of Id2 adjacent to the C-terminus bound to Akt (*Figure 1E*).

To determine whether Id2 is a substrate of Akt kinase, we generated constructs of GST-tagged Id1-Id4, as four *Id* genes with highly conserved HLH regions have been identified in human cells (*Figure 1—figure supplement 2A*), and performed in vitro kinase assays with purified active Akt protein. Among the Id family proteins, only the Id2 protein was substantially phosphorylated by active Akt, although all Id 1–4 proteins interacted with Akt (*Figure 1F*, *Figure 1—figure supplement 2B*). We verified that the specific site of Id2 phosphorylation by Akt is located on the very end of the N-terminus, within amino acid residues 1–37 (*Figure 1G*). Employing anti-phospho-Ser/Thr Akt substrate sequence antibody, we supported the notion that Id2 has a putative phosphorylation site for Akt (*Figure 1—figure supplement 2C*).

In vitro kinase assay with phospho-ablated mutant forms of Id2, revealed that Id2-S14A completely lacked phosphorylation, whereas WT-Id2 and mutation on Serine 5 of Id2, which has been shown to be phosphorylated by cyclin A/cdk2 (*Hara et al., 1994*), showed strong phosphorylation by Akt. This finding was in agreement with our phospho-proteomic analysis, which revealed S14 to be a putative phosphorylation site of Akt (*Figure 1H*, *Figure 1—figure supplement 2D*). To confirm specific phosphorylation at S14 on Id2, we generated phospho-specific antibody that recognized S14 (*Figure 1—figure supplement 2E*) and demonstrated that S14 is indeed phosphorylated in primary cultured neurons as they develop (*Figure 1I*). Taken together our data demonstrated that Id2 is a novel binding partner and kinase substrate of Akt in the developing neuron.

## Akt controls Id2 protein stability in the neuron

Id2 was highly expressed in the mouse hippocampus in the embryonic stages (E14 and E17) and decreased after birth and overtime (P7-P28). The level of phospho-Akt paralleled the decrease in Id2, showing a drastic decrease after P14 in the postnatal hippocampus of mouse brain (*Figure 2A*). Only the level of Akt1, but not that of Akt2 or Akt3, was reduced in a time frame similar to that of Id2 (*Figure 2B*), suggesting that Akt1 might be relevant in the control of Id2 protein level in neurons, correlating with our observation that Akt1 specifically interacted with Id2 (*Figure 1C*). Based on this finding, we focused our investigation on the biological significance of this interaction using Akt1, unless otherwise specified.

Id2 degradation in neurons is facilitated by APC/C$^{Cdh1}$, which inhibits axonal growth and Cdh1 is a regulatory subunit of the E3 ubiquitin ligase APC/C$^{Cdh1}$ responsible for Id2 degradation (*Lasorella et al., 2006*). Based on our finding of a development-dependent decline in Akt/Id2 signaling, we wondered if Akt activation regulates Id2 stability by blocking its proteasomal degradation. Treatment with the proteasomal inhibitor MG132 protected against Id2 degradation, confirming that the reduction in Id2 level is facilitated by the ubiquitin-proteasome system (UPS)-dependent degradation (*Figure 2—figure supplement 1*). Overexpression of Cdh1 markedly reduced endogenous Id2 level; importantly, this effect was prevented by Akt expression in PC12 cells (*Figure 2C*). In cortical neurons, Id2 protein level was proportionally increased with increased Akt level in the presence of Cdh1, indicating that Akt prevents Id2 degradation (*Figure 2D*). Accordingly, polyubiquitination of Id2 was efficiently abrogated in the presence, but not in the absence of Akt (*Figure 2E*), indicating that Akt regulates Id2 protein stability by preventing APC/C$^{Cdh1}$-mediated degradation.

Id2 was found to be associated with Cdh1. However, the interaction between Id2 and Cdh1 weakened with increased Akt expression, whereas the interaction of Id2 with Akt increased, suggesting that Akt competes with Cdh1 to bind Id2 (*Figure 2F*). The half-life of Id2 was lower in a phospho-ablated mutant (GFP-S14A) that could not be phosphorylated by Akt than in GFP-Id2-WT-expressing cells, whereas a phospho-mimetic mutant (GFP-S14D) showed more stable expression after cyclo-heximide (CHX) treatment (*Figure 2G*). This implies that Akt protects Id2 from Cdh1-mediated proteasomal degradation through phosphorylation of Id2.

To further verify the importance of Id2 phosphorylation by Akt for protein stability, we introduced a phospho-ablated mutant or phospho-mimetic mutant of Id2, along with HA-Cdh1, into PC12 cells. While Id2-WT and Id2-S14D rarely bound to Cdh1, the phospho-ablated mutant form of Id2 largely showed enhanced association with Cdh1, reflecting its instability and weak detection (*Figure 2H*). Moreover, Id2-WT was found to be more stable in the presence of CA-Akt than in that of KD-Akt, while the Id2-S14A mutant was not stabilized by CA-Akt as it could not be phosphorylated. The protein level of phospho-mimetic Id2-S14D was highly stable regardless of whether CA-Akt or KD-Akt was expressed (*Figure 2I*). Furthermore, using Akt inhibitor VIII, a chemical inhibitor of Akt signaling, we showed a reduction of Id2 protein levels as its phosphorylation is decreased upon inhibition of

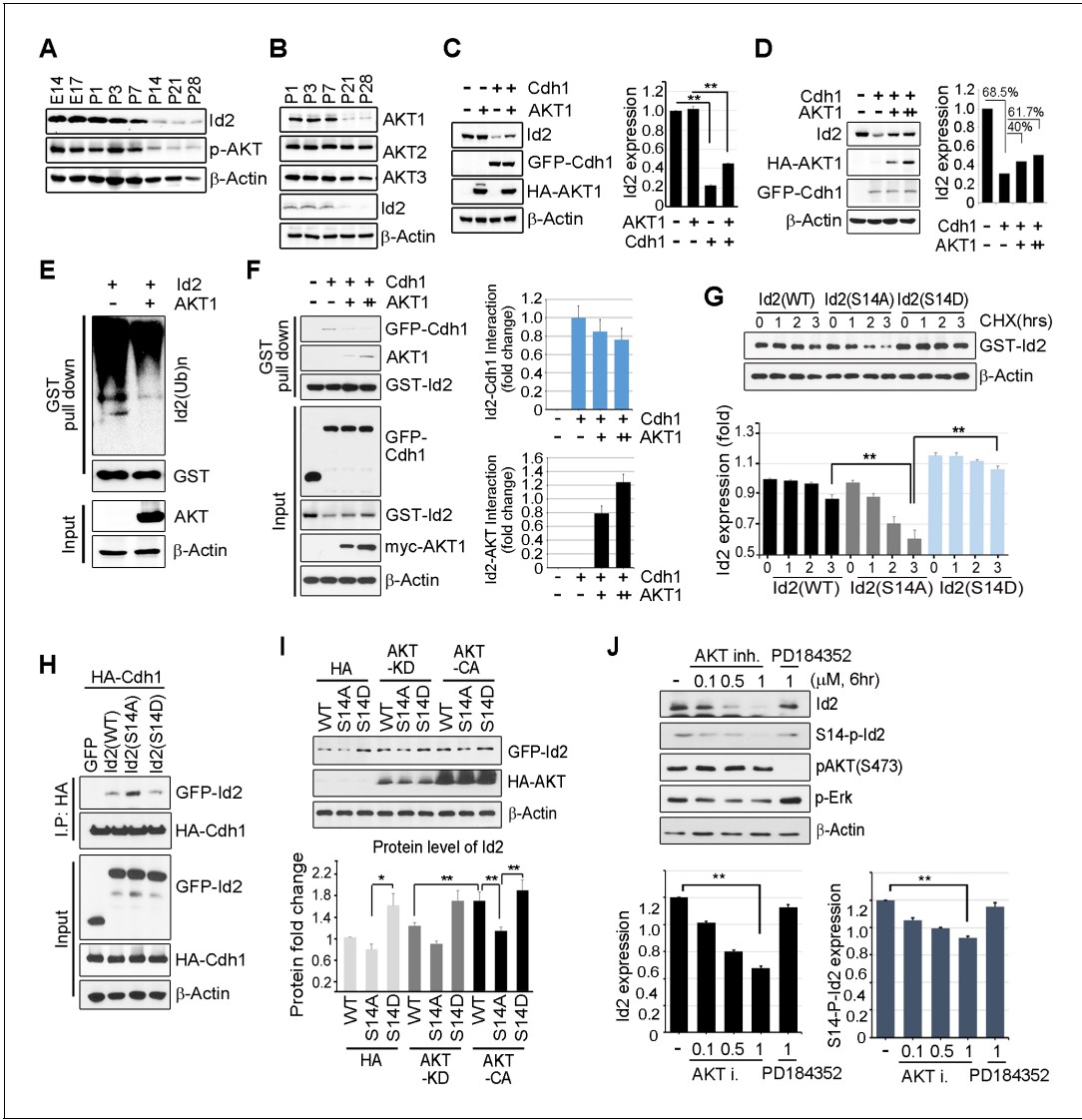

**Figure 2.** Akt controls Id2 protein stability in the neuron. (**A–B**) Lysates from mouse hippocampus of the indicated days were subjected to IB with the indicated antibodies. (**C**) PC12 cell were transfected with the indicated combination of HA-Akt or GFP-Cdh1 and the protein level was determined by IB (left). Densitometry analysis of IB is shown on the right. (**D**) PC12 cells were transfected with GFP-Cdh1 together with HA-vector or HA-Akt (+: 2 μg or ++:4 μg) and probed on IB (left). Densitometry analysis of IB is shown on the right. (**E**) GST-Id2 was co-transfected with HA-Akt into PC12 cells. Twenty-four hours after transfection, the cells were treated with the proteasome inhibitor MG132. GST-pull down assay was performed to determine ubiquitinated Id2. (**F**) PC12 cells were transfected with GST-Id2, GFP-cdh1 and increasing amounts of myc-Akt (+: 2 μg/++:4 μg) and the cell lysates were subject to GST pull-down. Immunoblot is shown on the left and quantification of the interaction affinity of GFP-cdh1 and GST-Id2 by densitometry analysis is shown on the right. (**G**) Transfected PC12 cells were treated with cycloheximide (CHX, 10 μM) as indicated time and probed on the IB (upper). Quantification of the Id2 protein levels by densitometry analysis (bottom). (**H**) HA-cdh1 was co-transfected with GFP-Id2 WT or mutants into 293T cells and protein levels of Id2 was detected by anti-GFP antibody after IP with HA antibody. (**I**) PC12 cells were transfected with GFP-Id2 WT, S14A, or S14D with HA-Akt KD or HA-Akt CA and probed on IB (left) Quantification of protein levels is shown in the bottom. (**J**) PC12 cells were treated with Akt inhibitor VIII (0, 0.1, 0.5 or 1 μM) or PD184352 (1 μM). Amounts of total and phosphorylated Id2 were determined by IB. *p<0.05, **p<0.005 versus indicated (**G** and **I**). Values in this figure represent mean ± SEM from three independent experiments and image shown here is representative from at least three independent experiments. See also *Figure 2—figure supplement 1*.

The following figure supplement is available for figure 2:

**Figure supplement 1.** Degradation of Id2 proteins by the ubiquitin-proteasome.

AKT phosphorylation, whereas in the presence of PD184352, a chemical inhibitor of MAPK signaling, Id2 stability or phosphorylation is not altered (*Figure 2J*). These data suggest that Akt-dependent Id2 phosphorylation enhances resistance to Cdh1-mediated degradation of Id2, interrupting the interaction between Id2 and Cdh1.

## Phosphorylation of Id2 by akt is essential for augmentation of axon growth and branching

Akt has been shown to be predominantly localized at the tip of the axon in developing hippocampal neurons (*Yan et al., 2006*). Furthermore, we found that the level of Akt protein and its activation state are closely related to the expression level of Id2 during development, and that Akt-mediated Id2 phosphorylation enhanced Id2 stability by preventing APC/C$^{Cdh1}$-mediated degradation; therefore, we wondered whether Akt regulates the function of Id2 in axonal growth. During differentiation of rat hippocampal neurons (up to in vitro day (DIV) 5), the spatial distributions of Id2 and Akt were visualized not only in the soma, but also prominently in the precursors of axons and dendrites in the early stages of differentiation (*Figure 3—figure supplement 1*, DIV1). Id2 and Akt were found in proximal axons, with expression tapering off along the distal axon (*Figure 3—figure supplement 1*, DIV2); however, the signals were strikingly intense at the distal part of the growing axon with a growth cone and axon branching points in later stages of axon growth (*Figure 3A*, *Figure 3—figure supplement 1*, DIV 4 - 5), suggesting that Id2 is potentially a downstream target of Akt in the regulation of axon growth and branching.

To delineate the roles of Akt/Id2 signaling in the regulation of axon growth, we transfected GFP-Id2 constructs into dissociated rat E18 hippocampal neurons and maintained them. Cultured hippocampal neurons at DIV three were immunolabeled with neuron specific class III beta tubulin (Tuj1) antibody to assess the extent of axon growth. Ectopic expression of Id2-WT led to considerably better axon growth and branching than that of control; S14D expressing neurons showed more abundant branching and extended length of axon than did the control or Id2-WT expressing neurons, revealing high expression at the tip of axon and branching. In contrast, the phospho-ablated mutant (S14A)-expressing neurons exhibited substantially shorter extent of axon growth and less branching in primary cultured hippocampal neurons (*Figure 3B,C*). Interestingly, we failed to detect GFP signal in the axonal tip of S14A expressing neurons despite no alteration of this signal in the soma, indicating lack of Id2 expression in the growth cone of the growing axon (*Figure 3B*, third panel). Taken together, these data imply that Akt regulates axon growth and branching by regulating Id2 phosphorylation and that this phosphorylation is essential for Id2 localization in the growth cone and branching.

## Akt regulates growth cone localization of Id2 in the developing neuron

As we found that Akt/Id2 accumulate in the axon tip and branching points in growing hippocampal neurons, and that Akt-mediated Id2 phosphorylation is essential for axonal growth and growth cone localization of Id2, we hypothesized that there might be spatial and temporal correlations between the expression and subcellular localization of Akt/Id2 with development of the growth cone. During the differentiation of hippocampal neurons, we found that prominent endogenous-Id2 expression occurred in the central domain of growth cone and partially colocalized with phalloidin-labeled F-actin, while S14-phospho-Id2 was localized on punctate structures all-around of the growth cone area and filopodia. S14-phospho-Id2 stained strongly in the growth cone leading edge and the peripheral domain, revealing notable co-distribution with phalloidin-labeled F-actin in the growing axon (*Figure 4A–C*). To more accurately determine the role of Akt/Id2 signaling in the growth cone, we monitored Id2 and S14-phospho-Id2 expression as neuronal development proceeded (*Figure 4D*, *Figure 4—figure supplement 1A*). In the early stage (stage 1: DIV 1) both Id2 and S14-phospho-Id2 were observed in the filopodial and lamellipodial structure of the leading margin. However, as the axon developed (stage II~III: DIV 2 and 3), Id2 was predominantly detected in the central microtubule-containing zone, demonstrating complete co-localization with beta tubulin, while S14-phospho-Id2 displayed a more intense signal at the peripheral-domain of growth cone, where there was relatively less beta tubulin staining (*Figure 4D,E*, *Figure 4—figure supplement 1A,B*). Stage determination was performed as previously described for hippocampal neurons (*Dotti et al., 1988*). Moreover, quantitative analysis that determined the expression level of Id2 or S14-phospho-Id2 from

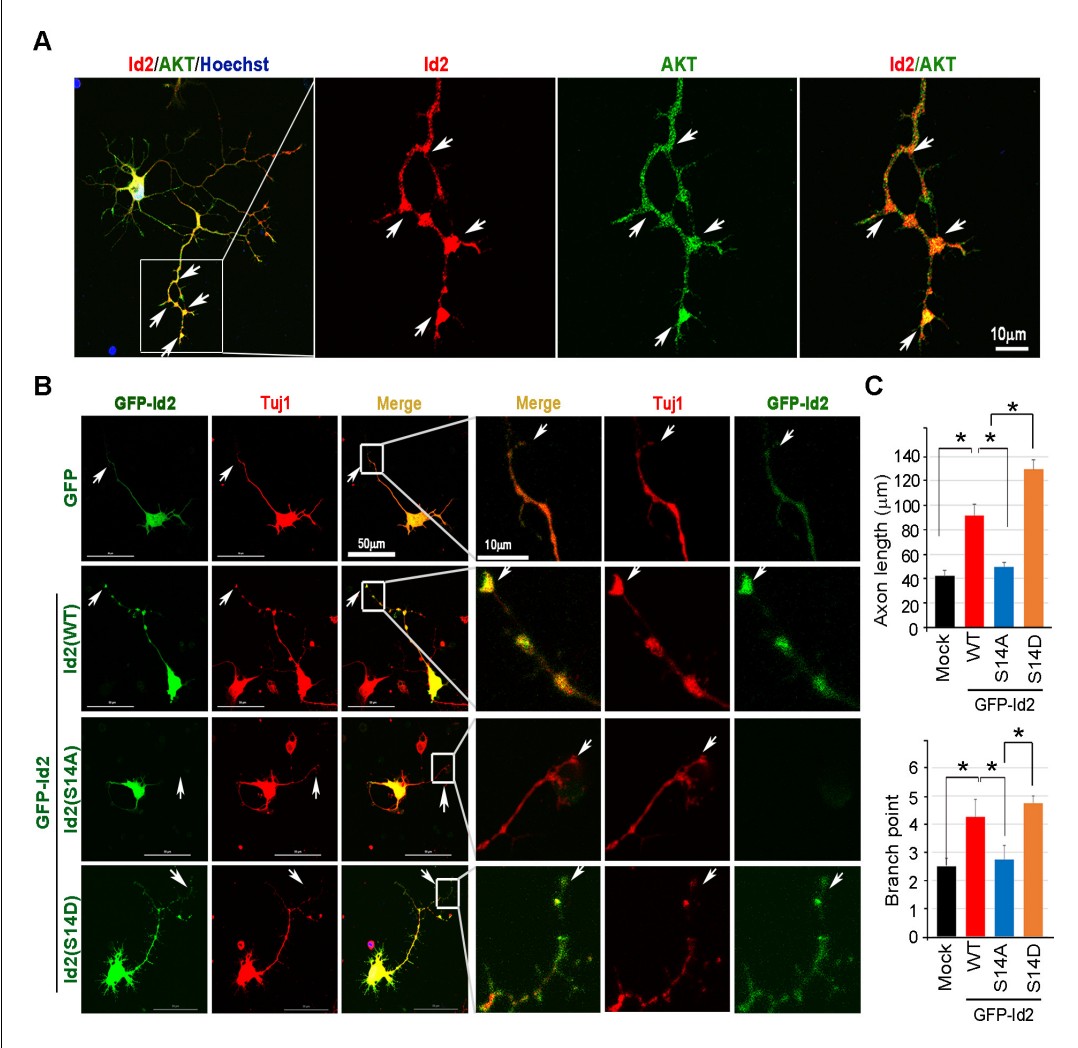

**Figure 3.** Phosphorylation of Id2 by Akt is essential for augmentation of axon growth and branching. (**A**) Representative merged image of localization of endogenous Akt and Id2 in the hippocampus neurons (DIV 4). The neurons stained for Id2 (red) and Akt (green). Right panel shows a higher magnification of the region indicated by a box. Scale bar, 10 μm. Image shown here is representative from at least three independent experiments. (**B–C**) Cultured neurons were transfected with GFP-Id2 WT, S14A, S14D or GFP vector control at day DIV one and fixed at DIV 3. Neurons were stained with anti-Tuj1(red). Representative images with a higher magnification of the region indicated by a box are shown in (**B**). Quantification of axon length and branching point measurements from three independent experiments is shown in (**C**). n = 16–24 cells. Error bars, SEM; Scale bar, 50 μm or 10 μm. *p<0.05 versus indicated. *Arrows indicate* axonal tip and branch points (A–B). See also *Figure 3—figure supplement 1*.

The following figure supplement is available for figure 3:

**Figure supplement 1.** Phosphorylation of Id2 by Akt is essential for augmentation of axon growth and branching.

soma to axonal tip, supported the notion that S14-phospho-Id2 is relatively enriched in the tip of the growth cone, with respect to that of Id2, as confirmed by fluorescence intensity analysis (*Figure 4F, G*).

To further confirm the specificity of anti-S14-phospho antibody, we depleted Id2 in the growing neuron using lentiviral-siRNA or inhibited Akt activity by chemical inhibitor of Akt. We selected Id2-siRNA that selectively reduced Id2 levels, but not the expression of unrelated protein such as beta-actin in PC12 cells (*Figure 4—figure supplement 2A*). Using selected Id2-siRNA, we generated and purified lentivirus that expresses Id2-siRNA and confirmed that GFP-lentiviral Id2-siRNA can specifically suppress endogenous Id2 protein expression by immunoblotting (*Figure 4—figure*

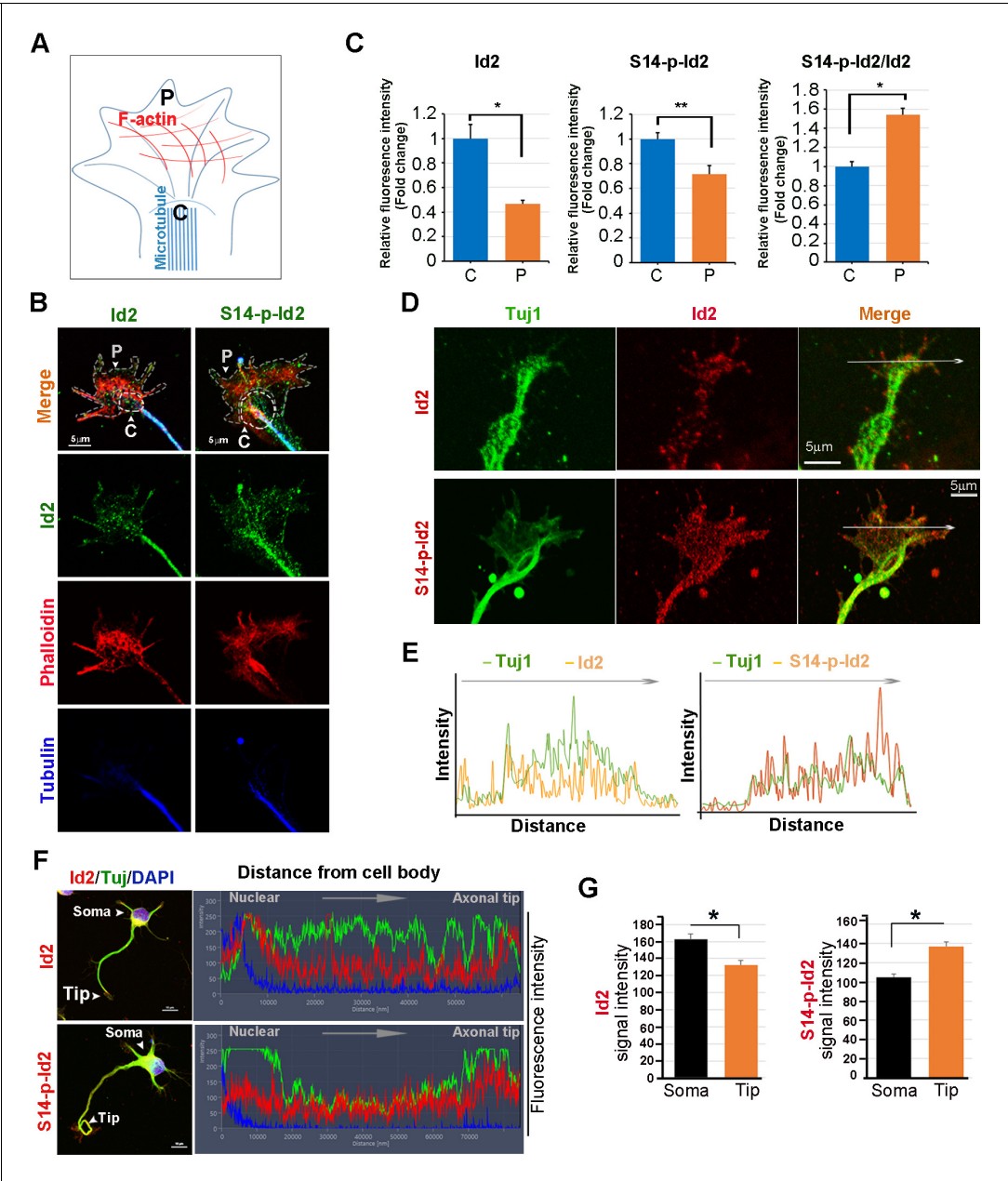

**Figure 4.** Akt regulates growth cone localization of Id2 in the developing neuron. (A) Schematic diagram of growth cone, showing microtubule mostly in the central [C] region and F-actin based peripheral [P]region. (B) Representative image of Id2 or S14-phospho-Id2 (green) with phalloidin labeled F-actin (red) and beta-tubulin (Tuj1:blue) in the growth cone of hippocampal neuron (stage3:DIV3). Arrows indicate example of [P]and [C] domain. Scale bar, 5 μm. (C) Quantification of S14-phospho-Id2/ Phalloidin or Id2/ Phalloidin at [P] and S14-phospho-Id2/Tuj1 or Id2/Tuj1 at [C] domain was averaged over multiple growth cones (right and middle). The ration of S14-phospho-Id2/Id2 at [C] and [P] was shown in left. n = 35. *p<0.05. **p<0.005. [P] or [C] domain is outlined by dashed gray or white line based on immunolabeling of phalloidin or Tuj1 in (B). (D) Representative image of beta-tubulin (Tuj1: green) with Id2 or S14-phospho-Id2 (red) in DIV2 neuron. Scale bar, 5 μm. The fluorescent image of DIV 1–3 is shown in *Figure 4—figure supplement 1A* and the original image of neuron for this representative growth cone is placed in *Figure 4—figure supplement 1B*. (E) Graphs plot the fluorescence intensity of immunolabeled Id2 (red) and Tuj1 (green) or phosphor Id2 (red) and Tuj1 (green) the arrowed line in *Figure 4D* is shown in each growth cone image. (F) The hippocampal neuron was fixed and stained with anti-Id2 or S14-phospho-Id2 antibodies (red). The neuron was stained with the Tuj1 (green), and nuclei were counterstained with DAPI. Scale bar, left: 20 μm. Relative immunofluorescence intensity profiles of Id2 and Tuj1 along the axon from cell body to axonal tip (right). (G) Quantification of Id2 and S14-phospho-Id2 signal intensity in the soma or axonal tip respectively. *p<0.05 versus control.Data represent mean ± SEM of three independent experiments. n = 20. See also *Figure 4—figure supplements 1* and *2*.

The following figure supplements are available for figure 4:

*Figure 4 continued on next page*

*Figure 4 continued*

**Figure supplement 1.** Id2 and S14-phospho-Id2 localized in the axonal growth cone of developing neuron.

**Figure supplement 2.** Knockdown of Id2 or inhibition of Akt impairs Id2 phosphorylation and its roles in the growth cone.

*supplement 2B*). Either knockdown of Id2 or treatment of Akt inhibitor diminished the specific signal of S14-phospho-Id2 in the growth cone with abnormal feature of growth cone (*Figure 4—figure supplement 2C–E*). Thus, our data indicate that S14 phosphorylation of Id2 probably drives its localization in the peripheral region of growth cone.

## Akt/Id2 signaling promotes axon growth by regulating growth cone development

We analyzed the localization of Id2 and S14-phospho-Id2 with phalloidin-labeled F-actin in the growth cone of neurons, we next asked whether Akt/Id2 signaling in the growth cone is involved in Id2 growth cone formation and function. ERM proteins link the actin cytoskeleton to the plasma membrane and play prominent roles in growth cone morphology and motility (*Mangeat et al., 1999*; *Dickson et al., 2002*). While ezrin and moesin expression is strongest in the central region of growth cone, radixin is highly stained in the peripheral region with phalloidin-label F-actin (*Marsick et al., 2012*). When we used an anti-ERM antibody that recognizes an epitope common to all ERM family members, our immunoprecipitation assay mouse brain extract (E18) showed that endogenous Id2 interacts with the ERM proteins (*Figure 5A*). Interestingly, compared to the binding affinity of Id2 to ERM, brain extracts immunoprecipitated with anti-S14-phospho-Id2 antibody showed a relatively strong interaction with the ERM proteins (*Figure 5B*), suggesting that S14-phospho-Id2 probably binds to radixin among the ERM proteins based on the peripheral distribution of S14-phospho-Id2 and radixin. Id2 was concentrated in the central region and in the base of some filopodia, while radixin expression was relatively abundant in the peripheral region (*Figure 5—figure supplement 1A*). Intriguingly, S14-phospho-Id2 was highly expressed in the peripheral filopodia, revealing co-distribution with radixin, as confirmed by fluorescence analysis (*Figure 5—figure supplement 1B*).

To further determine the importance of S14 phosphorylation on Id2 in the binding with radixin, we conducted an in vitro binding assay that demonstrated that Id2-WT interacted with radixin, whereas S14A failed to interact with GST-radixin (*Figure 5C*) and depletion of Akt by shRNA reduced the interaction between Id2 and radixin (*Figure 5D*, *Figure 5—figure supplement 1C–D*). Hence, Akt-mediated phosphorylation of Id2 is crucial for its association with radixin in the growing axon, which probably contributes to the growth cone function.

To determine the functional consequence of Akt/Id2 signaling in the growth cone, we utilized siRNA mediated knockdown of endogenous Id2. We infected the growing axon from hippocampal neuron (stage III) with either GFP-lentiviral control or GFP-lentiviral Id2-siRNA. Control GFP-expressing lentivirus-infected neurons showed normal growth cone architecture that was visualized with expression of radixin. However, knockdown of Id2 in the growing axon dramatically disrupted growth cone shape and cytoskeletal organization detected in neurons with reduced radixin levels (*Figure 5E*). Quantitative analysis of neuronal response involving alteration of the growth cone indicated noticeable reduction of growth cone size and shortening of axon length, implicating the importance of Id2 expression in the growth cone formation (*Figure 5F*). Accordingly, restoring Id2 signaling in the absence of endogenous Id2 expressing WT or S14D-Id2 rescued the deregulated axon growth. Conversely, RFP-vector control or S14A-Id2 mutant expression failed to rescue the effect of Id2 in the growth cone (*Figure 5G,H*).

We evaluated the effect of the ablation of S14 phosphorylation on Id2 in the growth cone. In the growth cone of the stage III neuron, radixin was evidently visualized in the filopodia of the growth cone and S14-phospho-Id2 was concentrated in axonal tip (*Figure 6A*). Upon treatment of the hippocampal neuron (stage III) with an Akt inhibitor, we found that the number of radixin expressing filopodia was greatly diminished in the axonal growth cone. We also observed that the intensity of radixin staining, as well as the intensity of S14-phospho-Id2 staining, as following the treatment, was

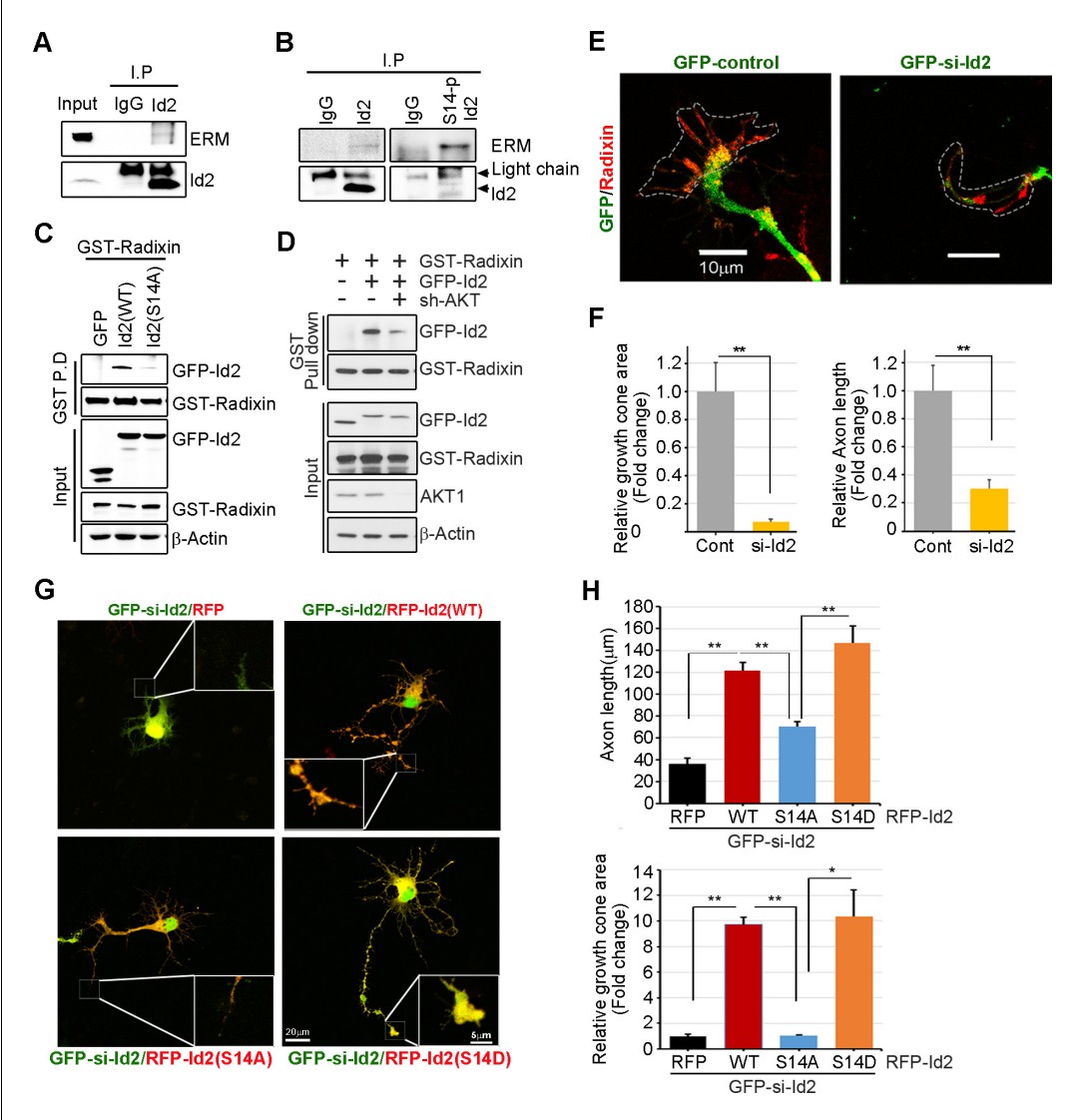

**Figure 5.** Akt/Id2 signaling promotes axon growth by regulating growth cone development. (**A–B**) E18 mouse brain lysates were subject to IP with anti-Id2 or anti-S14-phospho-antibody, followed by IB with anti-ERM antibody. (**C–D**) GST pull-down assay using cell lysates of PC12 cells transfected with indicated constructs following by IB. (**E**) Hippocampal neurons were infected with lenti-GFP- si-Id2 or lenti-GFP-scramble control at stage three and fixed after 48 hr. Neurons were stained with anti-radixin antibody (red). Growth cone area is outlined by dashed line based on immunolabeling of radixin. (**F**) Quantification of growth cone size and number of axonal length was based on radixin fluorescence from three experiments (n = 29–50). Scale bar, 10 µm. Error bars, SEM; **p<0.005 versus control. (**G** and **H**) GFP- si-Id2 was introduced to hippocampal neurons at DIV1 along with a series of RFP-Id2-WT, Id2-S14A or Id2-S14D and determined axon length and growth cone size at DIV4. Enlargement of growth cone area was shown in inserted box. (**H**) Quantification of axonal length and growth cone area (n = 15–21). Scale bar, 20 µm. Error bars, SEM; *p<0.05 **p<0.005. See also *Figure 5—figure supplement 1*.

The following figure supplement is available for figure 5:

**Figure supplement 1.** Akt/Id2 signaling promotes axon growth by regulating growth cone development.

remarkably lower than that of the control (*Figure 6A,B*). This indicates that S14 phosphorylation is critical for the proper function of radixin in the growth cone. In addition, knockdown of radixin in the growing neuron induced alteration in growth cone morphology and size and reduced the number of filopodia. This phenomenon parallels the alteration of endogenous S14-phospho-Id2 distribution as a reduction of growth cone (*Figure 6C*, *Figure 6—figure supplement 1*). However, ectopic

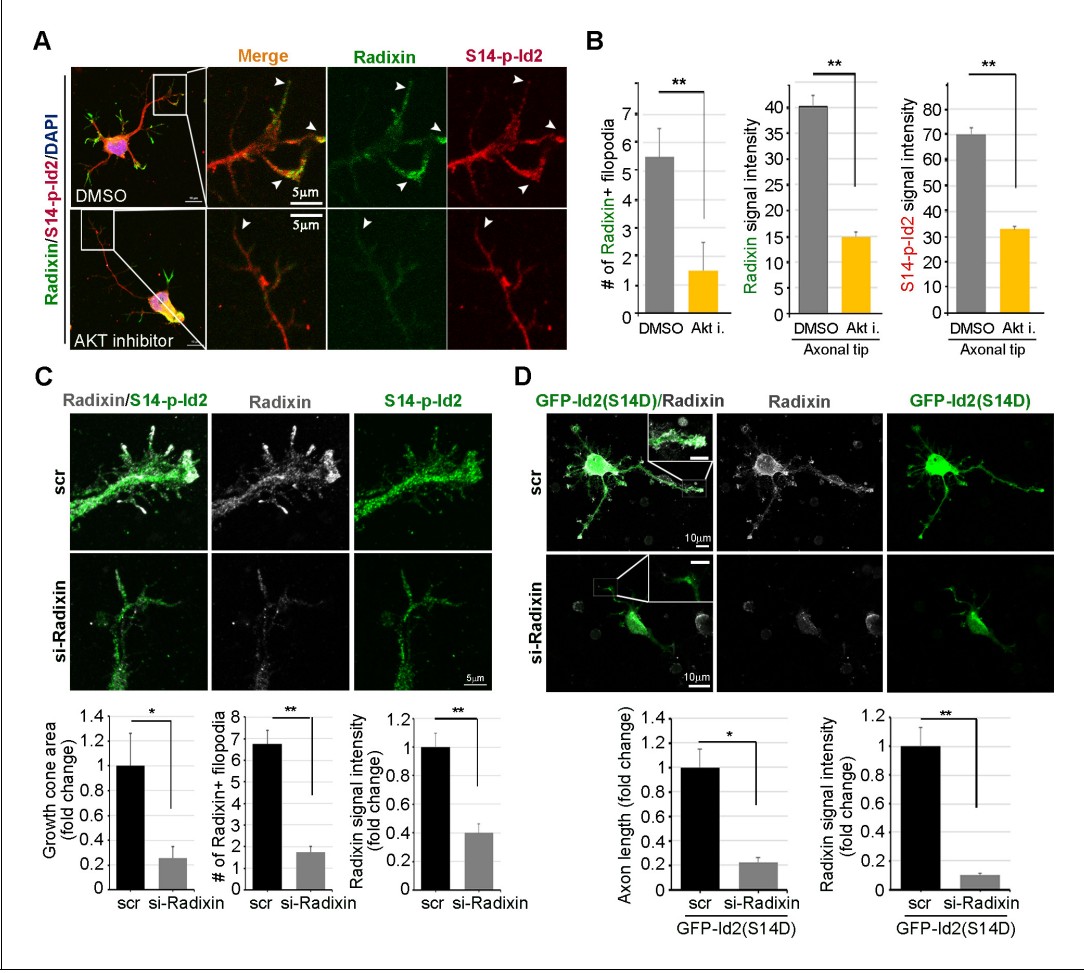

**Figure 6.** Akt/Id2 signaling is critical for the proper function of radixin in the growth cone. (A and B) Hippocampal neurons were treated with DMSO or AKT inhibitor for 4 hr. Enlargement of boxed area is in right. Scale bar, 50 μm or 5 μm. Arrows indicates radixin positive filopodia at the axonal tip. Bar graph shows radixin positive filopodia numbers in the axon (H, left) and relative radixin or S14-phospho-Id2 signal in the growth cone (n = 18–32) (H, middle and right). Error bars, SEM; **p<0.005 versus DMSO treated cell. (C) Hippocampal neurons were transfected with si-radixin or si-scramble control at stage three and fixed after 48 hr. Representative image (upper) and quantification of growth cone size, number of filopodia, and intensity of radixin were shown as bar graphs. scale bar, 5 μm. *p<0.05, **p<0.005 versus control. (n=15~18) (D) Hippocampal neurons were transfected with si-radixin or si-scramble and GFP-Id2 (S14D) at stage three and fixed after 48 hr. Enlargement of growth cone was shown in inserted box and bar graphs show axon length and radixin signal intensity. Scale bar, 10 μm. *p<0.05, **p<0.005 versus control. (n=17~20) See also *Figure 6—figure supplements 1–3*.

The following figure supplements are available for figure 6:

**Figure supplement 1.** Selection of si-RNA for radixin.

**Figure supplement 2.** Akt-mediated S14 phosphorylation is dispensable for the transcriptional repression activity of Id2.

**Figure supplement 3.** Akt-mediated S14 phosphorylation on Id2 is critical for its role in the growth cone.

expression of GFP-S14D-Id2 in neurons lacking radixin was not able to concretely maintain growth cone size and numbers of process (*Figure 6D*), underscoring the functional relevance of the Akt-Id2-radixin pathway for growth cone formation.

To verify specific roles Id2 in the axonal growth cone, we have further shown that Akt-mediated Id2 phosphorylation did not enhance transcriptional repression function of Id2 against E47. Akt-mediated S14 phosphorylation did not alter the interaction between Id2 and E47 and both S14D

and S14A mutants resulted in a repression of E47-activated luciferase activity similar to Id2-WT (*Figure 6—figure supplement 2A,B*). Moreover, employing Id2 constructs that cannot bind to E47 to antagonize transcriptional repression activity of Id2 showed that disruption of the HLH domain of Id1 by a substitution of proline to serine at position 74 or valine to proline at position 91 abolished its ability to inhibit E47 DNA binding (*Pesce and Benezra, 1993*). By substituting proline with serine at position 51, or valine with proline at position 68 in the Id2 counterpart, (*Figure 6—figure supplement 3A,B*), we demonstrated that although Id2 mutants (S51P or V68P) barely repress E47 transcriptional activity while Id2 WT successfully represses E47-mediated transcription displaying relatively high luciferase activity and the mRNA expression of Nogo receptor, one of well-known downstream gene of E47 (*Figure 6—figure supplement 3C,D*), Akt-mediated phosphorylation mimetic mutation in either S51P (S14D/ S51P) or V68P (S14D/V68P) provoked axon growth and growth cone localization of these mutants (S14D/ S51P or S14D/V68P), and its effect resembled that of Id2 WT in the growth cone (*Figure 6—figure supplement 3E,F*), suggesting that Akt mediated S14 phosphorylation probably regulates Id2 apart from transcriptional repression regulation. Taken together, these data suggest that in addition to transcriptional repression of axonal inhibitory molecules in the nucleus (*Perk et al., 2005*), Id2 expression in the growing axonal growth cone is a decisive factor for normal growth cone formation and axon growth. These data also suggest that Akt may not simply regulates transcriptional repression of Id2, but also growth cone localization of Id2 that is crucial for its association with radixin in the growing axon, an association that contributes to growth cone function.

## Akt/Id2 signaling contributes to axon regeneration in injured hippocampal slices

Because we found that Akt/Id2 signaling is critical for axon growth and growth cone formation in developing neurons, but that this signaling declines after birth, we tested whether reinstatement of the intrinsic growth ability of neuron by introduction of Akt/Id2 signaling contributes to axon regeneration in postnatal hippocampus after injury. We generated AAV2-Id2 and phospho-ablated/mimetic mutants of Id2 (*Figure 7—figure supplement 1A*) and employed an entorhinal-hippocampus (EH) organotypic slice co-culture (OSC) method that results in well preserved cytoarchitecture, closely reflecting the corresponding maturation schedule in vivo and known as an effective method for the study of axon regeneration (*Perk et al., 2005*; *del Río and Soriano, 2010*) (*Figure 7A*, *Figure 7—figure supplement 1B*). In agreement with previous studies (*Del Río et al., 1997*), very few short axons regrew in control virus-infected slices after injury (*Figure 7B* left). However, both the number and length of the regenerating axons that entered the hippocampus were greatly increased in the AAV2-Id2-WT-infected slices, whereas expression of the AAV2-Id2-S14A in the slices resulted in an approximate five-fold decrease in the number of regenerating axons growing into the denervated hippocampus from that of the Id2-WT (*Figure 7B,C*). Notably, Id2-S14D infection of the damaged brain slice substantially promoted axon regrowth at an even higher rate than that in the Id2-WT (*Figure 7—figure supplement 1C,D*). These results demonstrate that forced expression of Id2 after injury could promote axon regeneration and that this process is controlled by phosphorylation.

Without reintroduction of Id2, infection with active Akt expressing adenovirus of the injured region was insufficient to promote notable regrowth of axons. However, co-infection of active Akt with AAV2-Id2-WT significantly enhanced the length and numbers of regenerating axons, suggesting that Id2 functions downstream of Akt, to control axon regrowth (*Figure 7D,E*). In agreement with this finding, applying Akt inhibitor in the presence of AAV2-Id2-WT led to substantially lower axon regrowth than observed in the AAV2-Id2-WT-expressing slice with no inhibitor (*Figure 7F,G*). Thus, our data suggest that reactivation of the Akt/Id2 pathway in injured CNS neurons may be a useful therapeutic approach for promoting axon regeneration.

## Discussion

Our study demonstrated the molecular basis of intrinsic axon growth regulation of Akt/Id2 signaling in developing neurons, and the potential role of Akt/Id2 in CNS axon regeneration. Akt-mediated S14 phosphorylation of Id2 augments protein stability of Id2 through interruption of the interaction between Id2 and Cdh1, thereby reducing ubiquitination and proteasomal degradation of Id2. In the growing axon, Id2 is highly expressed at the branching point and axonal tip, leading to enhanced

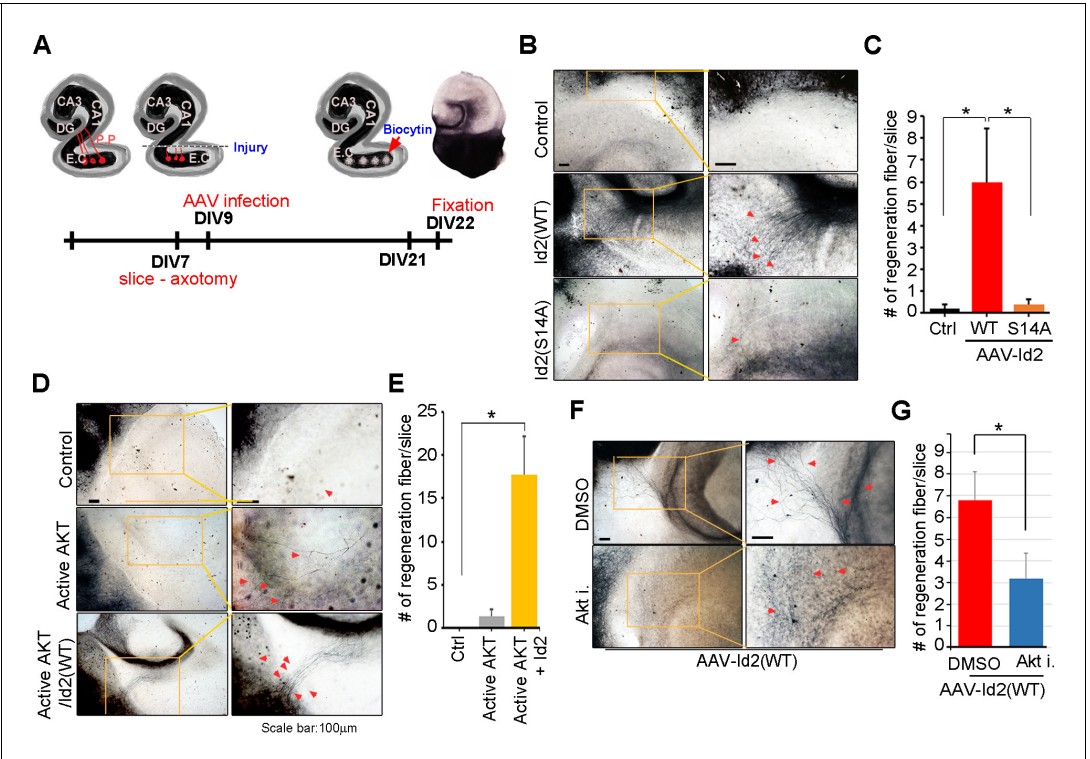

**Figure 7.** Akt/Id2 signaling contributes to axon regeneration in injured hippocampus slices. (A) Diagram illustrating experimental time course. The hippocampal slices were axotomized at the perforant path (PP) after DIV 7, and a series of AAV2- Id2 (WT, S14A, or S14D) or control with or without active Akt expressing virus, were infected on DIV 9. Slices were cultured for an additional 14 days. The anterograde axonal tracer biocytin was placed on the entorhinal cortex. Biocytin was visualized using the ABC-DAB method. (B–E) Representative images of biocytin tracing are in (B and D) and quantification for numbers of regenerating axon is in (C and E). *p<0.05. (n=36~45) Red arrows indicate the regenerated fibers. (F and G) Axotomized hippocampal slices were infected with AAV2-Id2 WT and treated with AKT inhibitor on DIV 9. Red arrows indicate the regenerated fibers. Scale bar, 100 μm. Images shown here is representative from at least three independent experiments and each value represents the mean ± SEM of triplicate measurements. *p<0.05 versus control. (n=36~42) See also *Figure 7—figure supplement 1*.

The following figure supplement is available for figure 7:

**Figure supplement 1.** Akt/Id2 signaling contributes to axon regeneration through growth cone formation.

axon growth and branching through S14 phosphorylation. More importantly, Akt/Id2 signaling is essential for growth cone formation, interacting with radixin at the peripheral region of the growth cone. Giving hope for translating intrinsic axon growth regulators into treatment in CNS injury, introduction of AAV2-Id2 at the injured EH dramatically enhanced axon regrowth and infection with AAV2-Id2 with active Akt noticeably enhancing axon regrowth, whereas AAV2-Id2 with Akt inhibitor failed to promote axon regeneration. This would suggest that reintroduction of Akt/Id2 signaling in injured CNS neurons might be a useful therapeutic approach for promoting axon regeneration.

During embryonic development, axonal growth occurs through a highly specialized mechanism in which the leading edge of the growth cone explores the environment and consolidates the cytoskeleton in the appropriate direction of growth (*Bentley and O'Connor, 1994*). Notably, the expression of Akt and Id2 occurs not only in the soma, where Id2 probably prevents expression of E47-induced anti-axonal genes (*Perk et al., 2005*), but also at the branching point and the tip of the growing axon, revealing accumulation of S14-phospho-Id2 at the peripheral region of the growth cone, including the actin-rich periphery and filopodia (*Figures 3* and *4*; *Figures 3—figure supplement 1* ; *Figures 4—figure supplement 1* ; *Figures 5—figure supplement 1* ). However, the role and regulation of Id2 in the growth cone has not been elucidated. Here we showed that ectopic expression of S14D-Id2 greatly enhances axon growth and branching, whereas S14A impairs axon growth and branching, rarely detecting growing distal axon (*Figure 3B*). Moreover, knockdown of Id2 from

developing neuron (stage III) that have distinguishable axons, elicited growth cone collapse and subsequent interruption of axon growth (*Figure 5E,F*). Furthermore, interference in Id2 phosphorylation by Akt inhibitor demolished axonal growth cone structure including filopodia (*Figure 6A,B*). Thus, these finding suggest that, in addition to its role as transcriptional repressor during development of the hippocampal neuron, Id2 acts as downstream mediator of Akt signaling, dictating its protein stability and positioning at the growth cone, and thereby contributing to the proper growth cone formation and axon growth. However, further investigation would be required to determine how S14 phosphorylation of Id2 by Akt steers Id2 localization to the growth cone, where Id2 interacts with radixin, which might be critical for normal growth cone architecture.

How might Akt/Id2 signaling regulate axon growth and growth cone formation? The roles of Akt in axon growth and growth cone establishment, proposed in multiple studies (*Dajas-Bailador et al., 2014*; *Gallo, 2008*; *Shi et al., 2003*): *Jiang et al., 2005*), occurs mostly through the regulation of GSK3β. However, an Akt-independent function of GSK3β in axon growth and polarization has also been suggested (*Shi et al., 2003*; *Diez et al., 2012*; *Zhang et al., 2014*), suggesting that other substrates of Akt may be implicated in axonal length regulation. On the other hand, mTORC1, another well-known downstream target of Akt, has been proposed to be another factor in axon formation (*Park et al., 2008*; *Morita and Sobue, 2009*; *Yang et al., 2014*). However, a clear link between Akt mediated regulation of mTORC1 and axonal growth in the CNS neuron has not been directly established. In this study, we proposed Id2 as a novel downstream effector of Akt signaling in the developing CNS neuron. Akt confined S14-phospho-Id2 at the tip of the growing axon thereby allowing its interaction with radixin, which functions in growth cone morphology and motility. We further demonstrated that ablation of S14 phosphorylation of Id2 led to a failure to bind to radixin and altered growth cone morphology (*Figure 5B–D* and *Figure 6A,B*), suggesting that, during neuronal development, Akt/Id2 signaling contributes to maintenance of the normal structure and functional organization of the growth cone, conceivably interacting with radixin, as S14D-Id2 failed to rescue the altered growth cone in the absence of radixin (*Figure 6C,D*). However, further investigation would be required to determine how S14 phosphorylation of Id2 by Akt steers Id2 localization to the growth cone, where Id2 interacts with radixin, which might be critical for normal growth cone architecture.

The ability of Id2 to promote axon growth and growth cone formation may be related to its ability to interact with components of the neuronal cytoskeletal machinery essential for establishment of neuron shape with a defined axon. We observed that phosphorylation by Akt, Id2 acts as an ERM binding protein, which bound F-actin and enriched the peripheral region of growth cone, while unphosphorylated Id2 predominantly localized in the central region of the growth cone where growing microtubule directed to actin network and, among Id family members (Id1-Id4), only Id2 possesses a SxIP motif-specific sequence to enable its interaction with end binding (EB) proteins and the most prominent microtubule (MT) plus end binding protein (+TIP), which promotes MT dynamicity and growth (*Coles and Bradke, 2015*); thus, it is possible to conjecture that Id2 may act as +TIP thereby capable to crosslink the two filaments, actin and microtubule, during axon growth. Two independent studies (*Arroyo et al., 2015*; *Huttlin et al., 2015*), have speculated that both Akt1 and Id2 are interacting partners of cytoplasmic linker associated protein 2 (CLASP2), which is known to act +TIP (*Beffert et al., 2012*). Moreover, in addition to radixin, Id2 also interacts with actin-associated enigma homolog (ENH) and is sequestered into the cytoplasm from nucleus during neural differentiation (*Lasorella and Iavarone, 2006*). Likewise, Id2 seems to be associated with both actin and microtubule filaments, and we also observed that Id2 forms a complex with EB1 and adenomatous polyposis coli (APC) (data not shown). We are currently planning experiments to determine whether Akt/Id2 signaling coordinates actin and microtubule dynamics in the growth cone.

Successful axon regeneration must be preceded by the successful formation of new growth cones. It has been reported that *Akt1* mRNA was markedly upregulated in injured neurons in adult rats, and that the phosphorylated Akt protein was dramatically increased after axotomy in adult motor neurons (*Namikawa et al., 2000*). A recent study has shown that infection with an Id2-adenovirus prevented E47-mediated induction of the Nogo receptor, which is responsible for three myelin proteins (myelin-associated glycoprotein, Nogo-A, and oligodendrocyte myelin glycoprotein) that account for most of the inhibitory activity of CNS myelin (*Silver and Miller, 2004*; *Lasorella et al., 2006*; *Schwab and Bartholdi, 1996*). In other studies, blocking growth inhibitory molecules allowed for limited regeneration (*Dimou et al., 2006*; *Fry et al., 2007*), suggesting the need for strategies

that enhance the intrinsic growth potential of CNS neurons to achieve regeneration. Based on our finding that the development-dependent decline in Akt/Id2 signaling (*Figure 2A*) and the reintroduction of Akt/Id2 signaling or enhanced Id2 stability by Akt-mediated phosphorylation are critical for axon growth and for the formation of a new growth cone, we attempted to determine the intrinsic growth potential contributed by Akt/Id2 signaling, when the growth ability of CNS neurons has declined. Intriguingly, AAV2-Id2 infection after injury in EH obviously facilitates axon regeneration (*Figure 7B,C*) and AAV2-S14D-Id2 or AAV2-Id2 with active Akt robustly enhanced axon regrowth (*Figure 7D,E*, *Figure 7—figure supplement 1C,D*), resulting in numerous axons entering the hippocampus after biocytin tracing. In contrast the functional blockade of Id2 by AAV2-phospho-ablated mutants, or treatment with Akt inhibitor in the presence of AAV2-Id2-WT, notably reduced axonal regeneration in the EH model (*Figure 7B,C,F,G*). Therefore, with the hope of translating intrinsic axon growth regulators into injured CNS, Akt/Id2 signaling could be a potent module to enhance the intrinsic growth potential in damaged CNS neurons

In summary (*Figure 8*), our study demonstrated that Id2 localization in the growing axonal tip of the growth cone, as a result of Akt-mediated phosphorylation, is a critical determinant of the intrinsic axonal growth ability in developing neurons, and manipulation of the growth control mechanism of

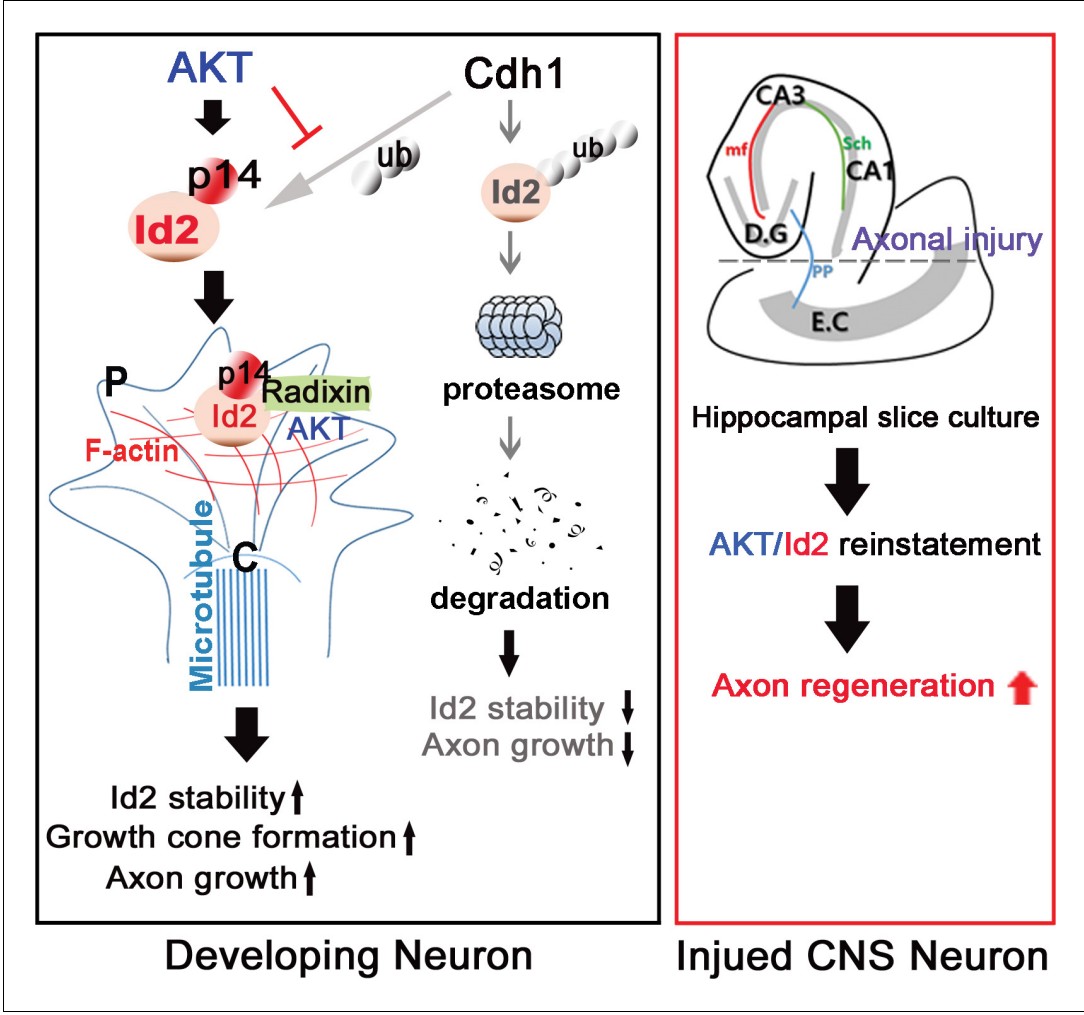

**Figure 8.** Schematic diagram of Akt/Id2 signaling pathway in the neuron. Akt-mediated S14 phosphorylation of Id2 augments its protein stability and growth cone localization, thereby promoting growth cone formation and axon growth in the developing neuron and contributing to axon regeneration in the damaged hippocampal slice culture.

Akt/Id2 signaling may provide new molecular targets for interventions to promote axon regeneration after CNS injury.

## Materials and methods

### Primary neuron and cell culture

The brains of E18 rat embryos were dissected, and hippocampi were removed and placed in a 15 ml tube with 14 ml of Hanks' Balanced Salt Solution (HBSS) on ice. The medium was carefully aspirated, leaving 2 ml of medium in the tube with the hippocampi. Papain (20 mg/ml in dissection medium) was added to digest the tissue. Samples were incubated for 20 min at 37°C. Digestion was stopped by washing the hippocampi twice with 4 ml of complete (10% FBS) medium. Then, 3 ml of Neurobasal media (NB, Invitrogen 21103–049) / B27 (Invitrogen 17504–044) was added, and the tissue was dissociated by gently triturating the hippocampi through a fire-polished Pasteur pipette. The cell mixture was diluted to 10 ml with NB/B27 and then filtered through a 40 or 70 μM strainer. Cells were spun at 1800 rpm for 5 min and suspended in 10 ml NB/B27. HEK293T cells and PC12 cells were cultured as previously described (*Ahn et al., 2004a*). HEK293T cells were cultured in Dulbecco's modified Eagle's medium supplemented with 10% fetal bovine serum (FBS) and 100 U of penicillin/streptomycin. PC12 cells were maintained in Dulbecco's modified Eagle's medium with 10% fetal bovine serum, 5% horse serum, and 100 U of penicillin/streptomycin at 37°C under a 5% $CO_2$ atmosphere. HEK293T cells (ATCCCRL-3216; PRID:CVCL_0063) and PC12 cells (ATCCCRL-1721; PRID:CVCL_0481) were obtained from ATCC. ATCC perform authentication and quality control tests on all distribution. All cell lines were authenticated by cell morphology monitoring, growth curve analysis, and mycoplasma detection using Mycoplasma detection kit (Roche) according to the ATCC cell line verification test recommendations periodically.

### Antibodies, siRNA, and chemicals

Anti-AKT1 (cat. 2938 s), -AKT2 (cat. 2964 s), -AKT3 (cat. 3788 s), -p-AKT (S473, cat. 4060 s), -ERM (cat. 3142 s), and -p-ERM (cat. 3141 s) antibodies were acquired from Cell Signaling (Danvers, MA, USA). Anti-GFP (cat. Sc-833s), -GST (cat. sc-138), -HA, -β-actin (cat. sc-47778), Anti-p-AKT (cat. Sc-514032), Id2 (cat. Sc- 398104) and -Id2 (cat. sc-489) antibodies were acquired from Santa Cruz Biotechnology (Dallas, TX, USA). Anti-FLAG (cat. F1804), -ezrin, -radixin, and -moesin antibodies were obtained from Sigma-Aldrich (St. Louis, MO, USA). Alexa Fluor 555 Phalloidin, Alexa Fluor 594 goat anti-rabbit and Alexa Fluor 488 goat anti-mouse secondary antibodies were obtained from Molecular Probes (Eugene, OR, USA). Anti-HSP70 was obtained from Abcam (Cambridge, MA, USA). siRNA (5'- GAGCUUAUGUCGAAUGAUAUU −3') for silencing of Id2 was obtained from Genolution (Republic of Korea). siRNA (sense 5'- ACAGUUGGUUUACGUGAGGUCUU −3', antisense 5'- GACC UCACGUAAACCAACUGUUU-3') for silencing of radixin was obtained from Genolution (Republic of Korea). For RT-PCR primer for Nogo receptor (NgR) were 5'- TATCCCCAGTGTTCCTGAGC-3' (forward) and 5'-GAGGTTGTTGGCAAACAGGT-3' (reverse) obtained from cosmogenetech (Republic of korea). Cycloheximide (CHX) was purchased from Duchefa Biochemie (Haarlem, Netherlands). MG132 was obtained from Sigma (St. Louis, MO, USA).

### Construction of recombinant DNA, AAV2, and lentivirus

A series of *Id2* (WT, S14A, S14D, P51S, V68P) were cloned into the pEGFP-C2 vector. The fragments of AKT were cloned into the pcDNA-GST vector. Radixin was cloned into the Kpn1-Not1 site of mammalian vector pcDNA-GST. For Expression and purification of GST fusion protein, various Id2 constructs were cloned into bacteria pGEX 4 T-1 vector. To generate AAV2 constructs, Id2 WT, S14A, and S14D were inserted into the AAV2-IRES-GFP vector and packaged in the 293 AAV2 cell line for production of high-titer AAV2 (Cell Biolabs, Inc., CA, USA). The AAV2 packaging service was provided by KIST (Korea Institute of Science and Technology, Seoul, Republic of Korea). Lentivirus purification was performed as previously described with few modifications (*Tiscornia et al., 2006*). pLenti-si Id2-GFP was packaged by cotransfection with the psPAX2 lentiviral packaging plasmid and the vesicular stomatitis virus envelope glycoprotein-expressing pMD2.G plasmid in 293T cells using the Neon Transfection System (Thermo Fisher Scientific Inc.). The culture supernatant was harvested after 72 hr, and the lentiviral particles were concentrated using a Beckman ultracentrifuge and a SW

41Ti rotor. The concentrated virus was resuspended in phosphate-buffered saline (PBS), aliquoted, and stored at −80°C.

## Co-immunoprecipitation and in vitro binding assays

For co-immunoprecipitation, cells were rinsed with phosphate-buffered saline (PBS) and lysed in buffer (50 mM Tris-Cl, pH 7.4, 150 mM NaCl, 1 mM EDTA, 0.5% Triton X-100, 1.5 mM $Na_3VO_4$, 50 mM sodium fluoride, 10 mM sodium pyrophosphate, 10 mM beta-glycerophosphate, 1 mM phenyl-methylsulfonyl fluoride (PMSF), and protease cocktail (Calbiochem, San Diego, CA)). Cell lysates (0.5 to 1 mg of protein) were mixed with primary antibody and protein A/G beads and incubated for 3 hr at 4°C with gentle agitation. The beads were then washed in lysis buffer and analyzed by immuno-blotting, as described above (*Lee et al., 2015*). For GST pull-down assays, cells were rinsed with PBS and lysed in buffer, as described above (*Zuo et al., 2015*; *Shin et al., 2015*). Cell lysates (0.5 to 1 mg of protein) were mixed with glutathione-sepharose beads and incubated for 3 hr at 4°C with gentle agitation. The beads were then washed in lysis buffer, mixed with 2x SDS sample buffer, boiled, and analyzed by immunoblotting.

## Immunofluorescence

Immunostaining was performed as described previously (*Choi et al., 2015*; *Kim et al., 2015*) with the following modifications. Cells grown on coverslips in 24-well plates were fixed in 4% paraformal-dehyde for 15 min, permeabilized in PBS containing 0.25% Triton X-100 for 10 min, and blocked in 1% BSA for 30 min. Cells were immunostained using primary antibodies and the appropriate Alexa Fluor 594 goat anti-rabbit and Alexa Fluor 488 goat anti-mouse secondary antibodies. Nuclei were counterstained with DAPI stain. Immunostained images were acquired using a laser scanning confocal microscope (LSM 710, Carl Zeiss, Germany). The confocal microscope was controlled using ZEN software and the acquisition was performed in the Research Core Facility, SBRI.

## In vitro kinase assay

In vitro Kinase assay was performed as previously described (*Ahn et al., 2004a*). Recombinant active Akt (Upstate Biotechnology) was incubated with ($1.8 \times 10^5$ Bq) $\gamma-32P$-ATP and 1 μg recombinant GST fusion protein in 30 μl kinase buffer (25 mM HEPES, 5 mM β-glycerophosphate, 10 mM MgCl2, 2 mM dithiothreitol, 0.1 mM NaVO3, and 200 μM ATP). Reactions were incubated at 30°C for 20 min and terminated by addition of Laemmli SDS sample dilution buffer. Proteins were separated by 10% SDS-PAGE, and phosphorylation was visualized by autoradiography.

## Silver staining

SDS-PAGE electrophoresis was performed with 15% gel. Gel was fixated with fixation solution (50% methanol, 12% acetic acid, 1/2000 of 37% formaldehyde) for 1 hr at room temperature. After fixation, gel was washed with 0.8 mM sodium thiosulfate for 2 min. Gel was incubated with silver nitrate solution for 15 min. After washing the gel in distilled water, the gel was developed developer solution (6% sodium carbonate, 0.016 mM thiosulfate, 0.05% formaldehyde) until the bands are visible.

## Luciferase activity assay

PC12 cells were transfected with promoter constructs and other DNAs by using Neon transfection system (Invitrogen). Cells were harvested 24 hr after transfection for luciferase assays. 20 μl of cell lysate containing 10 μg of protein was analyzed by using the luciferase assay system according to the manufacturer's instructions (Promega, Madison, WI, USA)

## Mouse hippocampal slice culture

Hippocampal slice cultures were prepared from P7 mouse brains. The 300-μm-thick brain slices were obtained by vibratome sectioning (Leica VT1200, Leica Biosystems) in chilled MEMp [50% (vol/vol) minimum essential medium (MEM), 25 mM HEPES, and 2 mM glutamine without antibiotics, adjusted to pH 7.2–7.3 with 1 M NaOH]. The slices were transferred onto semi-porous membrane inserts (Millipore, 0.4 μm pore diameter, Schwalbach, Germany). Intact slices were cultured at 37°C and 5% $CO_2$ in a standard medium MEMi [50% (vol/vol) MEM, 25 mM HEPES, 25% (vol/vol) HBSS, 25% (vol/vol) heat-inactivated horse serum, 2 mM glutamine, 1 ml of penicillin/streptomycin solution,

and 0.044% (vol/vol) NaHCO₃, adjusted to pH 7.2–7.3 with 1 M NaOH.] The medium was changed every other day. The hippocampal slices were axotomized after DIV 7, and the previously described AAV2s were used for infection at DIV 9. Slices were cultured for an additional 14 days. Anterograde axonal tracer of biocytin was placed on the entorhinal cortex at DIV 21. Hippocampal slices were fixed with 4% PFA at DIV 22. Biocytin was visualized using the ABC-DAB method.

## Statistical analysis

Data are expressed as mean ± SEM of triplicate measurements from three independent experiments. Statistical analysis was performed using Sigmaplot Statistical Analysis Software (Systat software, San Jose, CA, USA). All studies were performed in a blinded manner. Statistical significance was defined by Student's t-test ($^*p < 0.05$; $^{**}p < 0.005$).

## Acknowledgements

We thank DrAzad Bonni (Harvard Medical School) for pGFP-Cdh1 plasmid and Dr Haeyoung Suh-Kim (Ajou University, School of Medicine) for luciferase reporter gene of 3x-E-box-luc. We also thank Dr Jong Sun Kang (Sungkyunkwan University School of Medicine) for helpful comments on the manuscript. This work was supported by a National Research Foundation of Korea (NRF) grant funded by the Korean government (MSIP) (NRF-2013R1A2A2A01005324). The authors declare no competing financial interests.

## Additional information

### Funding

| Funder | Grant reference number | Author |
| --- | --- | --- |
| National Research Foundation of Korea | NRF-2013R1A2A2A01005324 | Jee-Yin Ahn |

The funders had no role in study design, data collection and interpretation, or the decision to submit the work for publication.

### Author contributions

HRK, J-YA, Conception and design, Acquisition of data, Analysis and interpretation of data, Drafting or revising the article; I-SK, IH, E-JJ, J-HS, Acquisition of data, Analysis and interpretation of data; AMB-M, RS, S-WC, K-HL, Acquisition of data, Contributed unpublished essential data or reagents

### Author ORCIDs

Raymond Swanson, http://orcid.org/0000-0002-3664-5359
Jee-Yin Ahn, http://orcid.org/0000-0003-0002-008X

### Ethics

Animal experimentation: This study was reviewed and approved by the Institutional Animal Care and Use Committee (IACUC) of Sungkyunkwan University School of Medicine (SUSM) (code16-21/16-22). SUSM is an Association for Assessment and Accreditation of Laboratory Animal Care International (AAALAC International; No. 001004) accredited facility and abide by the Institute of Laboratory Animal Resources (ILAR) guide. All experimental procedures were carried out in accordance with the regulations of the IACUC guideline of Sungkyunkwan University

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
