## [Decision Letter]

Thank you for submitting your article "Akt1-Id2 signaling is essential for growth cone formation and axon growth and promotes CNS axon regeneration" for consideration by *eLife*. Your article has been favorably evaluated by K VijayRaghavan (Senior Editor) and three reviewers, one of whom is a member of our Board of Reviewing Editors. The following individuals involved in review of your submission have agreed to reveal their identity: Keqiang Ye (Reviewer #2); Joseph W Lewcock (Reviewer #3).

The reviewers have discussed the reviews with one another and the Reviewing Editor has drafted this decision to help you prepare a revised submission.

Summary:

In this manuscript, Ko and colleagues argue for a new function for the transcription factor Id2 within the growth cone which is regulated via phosphorylation by the well-known kinase Akt. First, the authors identify Id2 (Inhibitor of DNA-binding protein 2) as a binding partner of Akt through mass spec. They then show that phosphorylation of Id2 on Serine 14 affects its stability, with consequences for axonal outgrowth and branching in primary hippocampal neurons. Next, they argue that Id2 is present not only in the nucleus but also in the growth cone. Finally, the authors use an in vitro hippocampal slice preparation of axon regeneration to demonstrate that activation of Id2 and Akt/Id2 promotes axon regeneration.

Essential revisions:

1) The manuscript contains some interesting findings linking Akt to Id2 signaling, with the consequences for regeneration. Unfortunately, the data surrounding the proposed function for Id2 growth cone are preliminary and are not adequate to support many of the authors arguments. The authors are challenged by the previously identified role for Id2-mediated transcription in axon outgrowth, and must demonstrate more than localization if Id2 protein in the growth cone in order to demonstrate a novel activity for axonal Id2. The co-localization and interaction studies in this study are not sufficient to rule out the possibility that Akt simply regulates Id2 mediated transcription, regardless of where this phosphorylation event may occur. Significant additional functional work would be required to define an axon specific role for Id2, for example with Id2 constructs that lack DNA binding activity.

2) We are concerned about the results described in Figure 4 where authors visualize phosphorylated forms of endogenous Id2 proteins in the growth cones of cultured neurons. However, there is no evidence to support that these signals indeed represent phosphorylated forms of Id2 proteins, or even a part of Id2 proteins. Perhaps the authors can use Id2 knockdown or some other methods to support the authenticity of the signals.

---

## [Author Response]

Essential revisions:

1) The manuscript contains some interesting findings linking Akt to Id2 signaling, with the consequences for regeneration. Unfortunately, the data surrounding the proposed function for Id2 growth cone are preliminary and are not adequate to support many of the authors arguments. The authors are challenged by the previously identified role for Id2-mediated transcription in axon outgrowth, and must demonstrate more than localization if Id2 protein in the growth cone in order to demonstrate a novel activity for axonal Id2. The co-localization and interaction studies in this study are not sufficient to rule out the possibility that Akt simply regulates Id2 mediated transcription, regardless of where this phosphorylation event may occur. Significant additional functional work would be required to define an axon specific role for Id2, for example with Id2 constructs that lack DNA binding activity.

We thank the reviewer for pointing out an important issue to address to strengthen our manuscript and for suggesting a relevant experiment. In the developing nervous system, Id2 enhances cell proliferation and inhibits the activity of neurogenic basic helix-loop-helix (bHLH) transcription factors and E proteins such as E47 and E12. Id2 does not bind to DNA because of the lack of a DNA binding domain, but it preferentially binds to ubiquitously expressed bHLH transcription factor and E proteins, and sequesters them from tissue-specific transcription factors such as NeuroD in processes such as neurogenesis, in which these other factors must compete with Id2 to heterodimerize with E proteins. Activation of the bHLH transcription factor E47 induces expression of a group of genes, including Nogo receptor, Sema3F, and Notch1 (Perk et al., 2005; Lasorella et al., 2006; Wang and Baker, 2015), which potently inhibit axon growth.

We have conducted a series of experiments to address the question the reviewer raised. In the revised manuscript (subsection “Akt/Id2 signaling promotes axon growth by regulating growth cone development”, fourth paragraph and fifth paragraphs), we described following new experimental data.

First, we asked whether Akt simply regulates Id2-dependent transcription. To determine this, we examined whether Akt-mediated Id2 phosphorylation enhances the interaction between Id2 and E47, thereby upregulating the inhibition of E47-mediated transcription. We co-transfected GFP- Id2-WT, S14D, or S14A with HA-E47 into PC12 cells and performed immunoprecipitation analysis. Id2 phosphorylation did not alter the interaction between Id2 and E47, showing that all Id2 constructs, regardless of phosphorylation status, had similar interactions with E47 (Figure 6—figure supplement 2). In addition, we performed luciferase reporter assays with 3 multimerized E-boxes driving the expression of luciferase (3xE-box-luc) (Jung et al., 2010). Co-transfection of PC12 cells (which express endogenous NeuroD) with 3xE-box-luc and E47 led to significantly higher luciferase activity than the background luciferase activity (3xE-box-luc alone) and, as expected, overexpression of GFP- Id2-WT notably reduced the luciferase activity. In accordance with the results of our binding analysis indicating that S14 phosphorylation of Id2 did not affect Id2 binding to E47, both S14D and S14A mutants also resulted in a repression of E47-activated luciferase activity, as similar to Id2-WT (Figure 6—figure supplement 2). Taken together, our results suggest that Akt-mediated S14 phosphorylation of Id2 is probably not necessary for E47 binding since dimerization in the bHLH family has been reported to be mediated by the HLH domain (Voronova and Baltimore, 1990; Pesce and Benezra, 1993) while Akt-dependent phosphorylation occurs N-terminal S14 residue. Therefore, Akt may not simply regulate Id2-mediated transcriptional inhibition but more likely is important in regulating axon growth and growth cone formation in the axonal tip.

Second, instead of using Id2 constructs that lack DNA binding activity due to lack of DNA binding ability of Id2, we generated Id2 constructs that cannot bind to E47 to antagonize transcriptional repression activity of Id2. In accordance with a previous observation that showed disruption of the HLH domain of Id1 by a proline-to-serine substitution at position 74 or a valine-to-proline substitution at position 91 abolished its ability to inhibit E47 DNA binding (Pesce and Benezra, 1993), we generated proline-to-serine substitutions at position 51 or valine-to-proline substitutions at position 68 in Id2 counterparts (Figure 6—figure supplement 3). Our binding assay showed that the interaction between E47 and P51S or V68P Id2 mutant was much weaker than that of Id2 WT (Figure 6—figure supplement 3). Moreover, both P51S and V68P mutations in the HLH domain of Id2 severely impaired the inhibition of E47-mediated transcription by Id2, revealing higher luciferase activity levels with P51S or V68P expression than observed for Id2-WT (Figure 6—figure supplement 3). Furthermore, co-transfection of Id2-WT with HA-E47 prevented the E47-mediated transcription of growth inhibitory molecules such as Nogo receptor, one of the best known inhibitors of axon growth (Schwab, 2004) whereas co-transfection of P51S or V68P Id2 mutant with E47 demonstrated relatively less inhibitory effect (Figure 6—figure supplement 3). To confirm the effect of Akt-mediated S14 phosphorylation on axon growth, we generated phosphor-mimetic P51S (S14D, P51S-Id2) or V68P Id2 (S14D, V68P-Id2) mutants and phosphor-ablated P51S (S14A, P51S-Id2) or V68P Id2 mutants (S14A, V68P-Id2), and determined axon growth and growth cone formation. Although both P51S and V68P Id2 mutant nominally repressed E47-mediated transcription, phosphor-mimetic mutation in P51S (S14D, P51S-Id2) or V68P Id2 (S14D, V68P-Id2) prominently promoted axon growth and led to growth cone localization of these mutant showing normal growth cone architecture resembling S14D-Id2 in the developing neuron (Figure 6—figure supplement 3). These data suggest that Akt-mediated S14 phosphorylation probably regulates Id2 independently from transcriptional repression regulation contributing normal growth cone formation and axon growth.

2) We are concerned about the results described in Figure 4 where authors visualize phosphorylated forms of endogenous Id2 proteins in the growth cones of cultured neurons. However, there is no evidence to support that these signals indeed represent phosphorylated forms of Id2 proteins, or even a part of Id2 proteins. Perhaps the authors can use Id2 knockdown or some other methods to support the authenticity of the signals.

In accordance with the reviewers’ suggestion, we introduced Id2-siRNA into cultured hippocampal neuron (DIV 3) and validated the specificity of the phosphor-S14 antibody. As shown in Figure 5, knockdown of Id2 disrupted growth cone architecture and was minimally detected in the growth cone while in the control, Id2 expression was prominent in the growth cone. Immunostaining against anti-S14-phospho antibody in the control SCR expressing neuron showed that the phosphor-S14 signal is co-localized with the Id2 signal in the growth cone, revealing an intense signal at the peripheral domain of the growth cone. However, we failed to detect any specific signal of anti-S14-phosphor antibody after knockdown of Id2 (Figure 4—figure supplement 2). In other ways, upon Akt inhibitor VIII treatment, S14 phosphor Id2 intensity was diminished following by decrease of Akt phosphorylation in the growth cone along with a reduction of growth cone area (Figure 4—figure supplement 2). We added this part in the revised manuscript (subsection “Akt regulates growth cone localization of Id2 in the developing neuron”, last paragraph). Moreover, upon Akt inhibitor VIII treatment, S14 phosphor Id2 was diminished and followed by decreased Akt phosphorylation along with a reduction of Id2 protein whereas upon MAPK inhibitor treatment, S14 phospho Id2 level was not altered following Akt phosphorylation, indicating anti-S14-phosphor antibody specifically detect Akt-mediated phosphorylated S14 residue of Id2 (Figure 2). Furthermore, depletion of Akt by shRNA-Akt also led to lower intensity of S14-phosphor Id2 than observed in the control vector transfected cells (Figure 5—figure supplement 1; subsection “Akt/Id2 signaling promotes axon growth by regulating growth cone development”, second paragraph). Taken together, anti-S14-phosphor antibody is suitable for detecting S14 phosphorylated Id2.